# Association between Proton Pump Inhibitor Use and Risk of Hepatocellular Carcinoma: A Korean Nationally Representative Cohort Study

**DOI:** 10.3390/jcm11102865

**Published:** 2022-05-19

**Authors:** Soungmun Kim, Seogsong Jeong, Sun Jae Park, Jooyoung Chang, Seulggie Choi, Yoosun Cho, Joseph C. Ahn, Gyeongsil Lee, Joung Sik Son, Sang Min Park

**Affiliations:** 1Graduate School of Data Science, Seoul National University, Seoul 08826, Korea; munkim97@gmail.com; 2Department of Biomedical Sciences, Seoul National University College of Medicine, Seoul 03080, Korea; sunjaepark.a@gmail.com (S.J.P.); joomyjoo@gmail.com (J.C.); seulggie@gmail.com (S.C.); gespino1.gs@gmail.com (G.L.); 3Department of Biomedical Informatics, CHA University School of Medicine, Seongnam 13488, Korea; seogsongjeong@gmail.com; 4Total Healthcare Center, Kangbuk Samsung Hospital, Sungkyunkwan University School of Medicine, Seoul 06351, Korea; misslonghorn46@gmail.com; 5Division of Gastroenterology and Hepatology, Mayo Clinic, Rochester, MN 55905, USA; ahn.joseph@mayo.edu; 6Estherformula Medical Food R&D Center, Seoul 06019, Korea; 7Department of Internal Medicine, Hallym University Sacred Heart Hospital, Anyang 14068, Korea; medical114@naver.com

**Keywords:** epidemiology, pharmacoepidemiology, proton pump inhibitor, hepatocellular carcinoma, liver cirrhosis, chronic liver diseases, primary liver cancer, defined daily dose, cohort study, health screening

## Abstract

(1) Background: The association between proton pump inhibitor (PPI) use and hepatocellular carcinoma (HCC) has been controversial, especially in the general population. We aimed to determine the impact of PPI on HCC risk in participants without liver cirrhosis or chronic hepatitis virus infection. (2) Methods: We assessed 406,057 participants from the Korean National Health Insurance Service database who underwent health screening from 2003 to 2006. We evaluated exposure to PPI before the index date using a standardized daily defined dose (DDD) system. The association of proton pump inhibitor use with the risk of HCC was evaluated using multivariable-adjusted Cox proportional hazards regression. (3) Results: Compared with non-users, PPI use was not associated with the HCC risk in low (<30 DDDs; aHR, 1.07; 95% CI, 0.91–1.27), intermediate (30 ≤ PPI < 60 DDDs; aHR, 0.96; 95% CI, 0.73–1.26), and high (≥60 DDDs; aHR, 0.86; 95% CI, 0.63–1.17) PPI groups in the final adjustment model. In addition, risks of cirrhosis-associated HCC and non-cirrhosis-associated HCC were not significantly associated with PPI use. The results remained consistent after excluding events that occurred within 1, 2, and 3 years to exclude pre-existing conditions that may be associated with the development of HCC. We also found no PPI-associated increase in HCC risk among the selected population, such as those with obesity, older age, and chronic liver diseases. (4) Conclusions: PPI use may not be associated with HCC risk regardless of the amount. We call for future studies conducted in other regions to generalize our findings.

## 1. Introduction

Primary liver cancer is the sixth most common cancer worldwide and the third leading cause of cancer-related mortality in 2020 [1]. Liver cancer incidence rates were highest in South-Eastern and Eastern Asian countries, of which the incidence of liver cancer was the highest in South Korea and Thailand [2]. Hepatocellular carcinoma (HCC) accounts for approximately 85–90% of all primary liver cancers as the dominant histological type in most countries [3]. HCC tends to have a poor prognosis because it is often diagnosed at the advanced stage compared to other cancers. In 2018, the incidence of HCC has been the seventh highest among all cancer types in South Korea and has not decreased over the past decade [4].

Hepatitis B virus (HBV) and hepatitis C virus (HCV) infections, diabetes mellitus (DM), obesity, aflatoxin, and alcohol consumption are known as risk factors for developing HCC [5,6,7]. Globally, HBV and HCV infections account for most cirrhosis and primary liver cancer. HBV accounts for at least 50% of cases of HCC as a major risk factor for HCC and HCV accounts for approximately 25% of cases of HCC [8,9]. Most HBV-related HCC (up to 70%) occurs in cirrhotic liver and the risk of HCC is 17-fold higher in infected participants with HCV compared to participants without HCV [10,11]. Obesity and DM, another risk factor for the development of HCC, increase the risk of HCC in participants without HBV and/or HCV infection; the risk of HCC in participants with these two conditions combined is 10-fold higher [6]. In addition, the incidence of HCC is high in areas where foods contaminated with aflatoxin, known as carcinogens, are common, and excessive alcohol consumption increases the risk of HCC through cirrhosis [7].

The proton pump inhibitor (PPI), first introduced in the late 1980s, is a medicine that inhibits gastric acid secretion and is used for the treatment of peptic ulcers, gastroesophageal reflux disease, and eradication of Helicobacter pylori [12,13]. PPI is generally known as a safe and effective medicine, but it has been pointed out that PPI use is associated with adverse outcomes, such as acute and chronic kidney disease, hypomagnesemia, and osteoporotic fractures [12,14]. In previous studies, results of the association between PPI use and the risk of liver cancer have been controversial [13,15]. In addition, PPI users with chronic liver disease (CLD) showed significantly increased HCC risk compared to non-users, but this association was not significant in Asian studies, mainly using Taiwan studies with strict PPI regulations [16]. Therefore, we conducted a retrospective cohort study with a large population to evaluate the association of HCC risk in PPI users in South Korea where PPI regulation is not strict. 

## 2. Materials and Methods

### 2.1. Study Population

The Korean National Health Insurance Service (NHIS) is a quasi-governmental organization that provides health insurance to all citizens. The NHIS collects health claims data for health administrative records of the entire population [17]. The NHIS database consists of sociodemographic information, inpatient and outpatient visitations and treatment, diagnoses, prescription of medicines, and health screening results. In addition, Korean adults aged 40 or older are provided with a biannual health screening examination and the participation rate in 2014 was 74.8%. The health screening examination database consists of a simple random sample of 10% of all health screening examination participants, and data can only be used anonymously for research purposes without the consent of individual patients under the National Health Insurance Act. The validity of the NHIS database is explained in detail elsewhere [17,18]. This study was conducted on 467,019 participants using health screening excitation data at the National Health Insurance Service-National Sample Cohort recorded between 2003 and 2006 (Figure 1). Participants with a history of hepatitis (*n* = 18,402), a history of cirrhosis (*n* = 8535), and a history of HCC (*n* = 577), which are risk factors for liver cancer, before the index date (1 January 2007) were excluded. We also excluded PPI users in 2002 (*n* = 3420) to minimize the impact of PPI use before the study period. Lastly, participants who have missing key variables were excluded (*n* = 30,028). The analysis was conducted with a total of 406,057 participants and was followed up from the index date until the diagnosis of HCC, cirrhosis, or until 31 December 2013, whichever came earliest. The Institutional Review Board of Seoul National University Hospital approved this study (E-2108-136-1246). The requirement for informed consent was waived since the NHIS-National Sample Cohort is provided for research purposes in an anonymized form according to the confidentiality guidelines.

### 2.2. Key Variables

The primary outcome was HCC, which was defined using the International Classification of Diseases, 10th Revision (ICD-10) code of C220. Chronic liver disease was defined as ICD-10, K70-K77. Cirrhosis-associated HCC was defined when a participant occurred liver cirrhosis (ICD-10, K703, and K746) within the follow-up period before the date of HCC.

All covariates and cumulative PPI usage information, which is our main exposure of interest, were extracted before the index date. PPI usage data were collected by each prescription recorded in the dataset such as dates, the daily dose, the number of pills, and the number of days supplied. We used the defined daily dose (DDD) recommended by the World Health Organization as a unit representing the amount of PPI prescribed. DDD is the average daily maintenance dose of a drug consumed by adults and the cumulative daily dose was calculated in units of DDD [19]. Based on the DDDs, PPI users were classified into three groups: low (<30 DDDs), intermediate (30–60 DDDs), and high (≥60 DDDs).

We included age (continuous; years), sex (categorical; men and women), household income (categorical; classified into quartiles), Charlson comorbidity index (categorical; 0,1, and ≥2), systolic blood pressure (continuous; mmHg), diastolic blood pressure (continuous; mmHg), BMI (continuous; kg/m^2^), fasting serum glucose (continuous; mg/dL), cigarette smoking (categorical; never, past, current), alcohol consumption (categorical; 0, 2–3/month, 1–2/week, 2–3/week, ≥5/week), and physical activity (categorical; 0, 1–2, 3–4, ≥5 time/week) in the analysis as covariates to adjust the potential confounding effects.

### 2.3. Statistical Analysis

Categorical and continuous variables were presented as *n* (%) and median (interquartile range; IQR). The crude rate was calculated by dividing the number of events by person-years, which was presented as event/10,000 person-years. In all analyses, non-PPI users were used as the reference groups to calculate the hazard ratio (HR) and 95% confidence interval (CI) using the Cox proportional hazards regression model. The Cox proportional hazards regression model was chosen to evaluate the association between PPI use and risk of HCC considering the longitudinal study design. The first model was analyzed without using adjustment variables, and the second model was analyzed by adjusting only age and sex variables. The final model was adjusted for age, sex, household income, BMI, systolic blood pressure, fasting serum glucose, cigarette smoking, physical activity, and Charlson comorbidity index (CCI).

We used cirrhosis-associated HCC and non-cirrhosis-associated HCC as the secondary outcomes to analyze the subdivided association using the above models. Sensitivity analysis was conducted by washing out events that occurred within 1, 2, and 3 years to confirm the consistency of the results. We also carried out stratified multivariable analyses according to demographic characteristics, aspirin use, non-steroidal anti-inflammatory drugs (NSAIDs) use, gastroesophageal reflux disease (GERD), and CLD to evaluate the interaction with PPI use and HCC risk. *p* = 0.05 was considered statistically significant for all analyses. All data mining and statistical analyses were performed using SAS Enterprise Guide 6.1 (SAS Institute Inc., Cary, NC, USA).

## 3. Results

Table 1 indicates the descriptive characteristics of the study population. The median age of the participants was 51 years (IQR, 45–60), with 53.1% of men and 46.9% of women having a similar sex ratio. The highest proportion of household income was 34% (*n* = 138,004), which was the fourth quartile. More than half of the participants were non-smokers (71.4%, *n* = 289,888) and non-drinkers (60.0%, *n* = 243,474), and almost half of the participants were physically inactive (52.6%, *n* = 213,675), respectively. 

The association of PPI use with the risk of HCC is shown in Table 2. Participants are classified into four groups based on PPI use and DDDs. During 2,772,507 person-years of follow-up, 1466 overall HCC cases, 282 cirrhosis-associated HCC cases, and 1184 non-cirrhosis-associated HCC cases were identified, respectively. Regardless of low, intermediate, and high DDDs, PPI use was all insignificant with overall HCC, cirrhosis-associated HCC, and non-cirrhosis-associated HCC compared to non-PPI users in the final adjustment model. Even ≥60 DDDs users showed that the use of PPI was not significantly associated with overall HCC (adjusted hazard ratio [aHR], 0.86; 0.63–1.17), cirrhosis-associated HCC (aHR, 1.20; 0.65–2.22), and non-cirrhosis-associated HCC (aHR, 0.78; 0.54–1.12).

Additionally, we conducted another analysis that classified PPI users into various risk groups based on tertiles, which supported the primary findings derived in DDD-stratified groups (Appendix A). The sensitivity analysis conducted by washing out for 1, 2, or 3 years also consistently showed that the use of PPI was not associated with HCC risk in accordance with the primary findings (Appendix A). 

In stratified analyses, some groups such as those <60 years old, men, and non-smokers showed decreased HCC risk as the DDDs of PPI usage increased without statistical significance (Figure 2A,B). There was no significant association between PPI use and HCC risk found for all other subgroups stratified according to household income, obesity, CCI, smoking, alcohol consumption, physical activity, GERD, aspirin, and non-steroidal anti-inflammatory drugs (Appendix A). Even ≥60 DDDs PPI users with CLD were not associated with HCC risk (aHR, 0.93; 0.61–1.41). In addition, all key variables showed no significant interaction. However, some groups showed decreased HCC risk as PPI doses increased without statistical significance. 

## 4. Discussion

We found that PPI use is not significantly associated with overall HCC, cirrhosis-associated HCC, and non-cirrhosis-associated HCC in this retrospective nationally representative cohort study, even among PPI users with CLD. Therefore, the use of PPI may not be restricted to the general population in terms of HCC risk. Future studies exploring a selected population who may be affected by the use of PPI are necessary to better confirm the safety of PPI against HCC risk.

Previous studies have shown the association between PPI use and cancer risks. Since the use of PPI can worsen astrophic gastritis [20], many previous studies were conducted on PPI use and the risk of gastric cancer. They found that PPI use is associated with an increase in gastric cancer, and long-term PPI use even after Helicobacter pylori eradication increased the risk of gastric cancer. An association between PPI use and gastric cancer was found, with HRs in the range of 1.45–2.44 [20,21]. Additionally, some previous studies have also shown the significant association of PPI use with the risk of colorectal cancer. The association of PPI use with colorectal cancer was shown with HRs in 1.06 elderly per year [22]. Several studies related to pancreatic cancer and PPI use have also been examined. PPI usage was associated with pancreatic cancer by showing OR in 1.75 and standardized incidence rate ratios (SIRs) in 2.22 [23,24].

Several studies have conducted meta-analyses and nested case–control studies to show the association between PPI use and liver cancer or HCC risk with conflicting results. Studies in the UK, Scotland, and Taiwan reported that PPI use was associated with an increase in HCC risk, while other studies conducted in Taiwan reported that PPI use was not significantly associated with an increase in HCC [13,15,25]. There are previous studies related to PPI use and HCC risk in CLD patients. Previous studies reported that PPI use is associated with an increased HCC risk in Western populations, but no such association has been found in Asian populations [16]. However, previous studies did not include confounding factors such as smoking and alcohol consumption [13,25], or the number of studies included was small [16]. Unlike these studies, our models were adjusted to sufficient confounding factors such as smoking, alcohol consumption, and systolic blood pressure. Additionally, our study used the health screening examination cohort and consistently set the index date of all participants as of 1 January 2007, which allowed concise evaluation of PPI use. Therefore, our results may provide further evidence in determining the association of PPI with HCC risk. 

Although no obvious pathological mechanism has been identified between PPI use and HCC risk, several potential mechanisms have been suggested. Reducing gastric acid using PPI can increase intestinal microbes and cause bacterial infection by gastrointestinal bacteria overgrowth [26,27]. This may cause toxicity, inflammation, and DNA damage to the bile duct and liver cells, which can cause HCC development [28]. In CLD patients, PPI toxicity may occur in the liver because PPI is metabolized in the liver and it leads to hypergastrinemia, which can cause hepatocarcinogenesis [29,30]. PPI use is metabolized in the liver by cytochrome CYP450 and secreted by kidneys. Renal insufficiency has almost no effect on PPI clearance while the cumulative risk of liver impairment increases due to increased drug half-life prolongation [31]. However, no association was found between increased HCC risk and PPI use in our study. Since PPI use may still trigger the development of HCC in individuals with liver dysfunction, we suggest not to overinterpret our study. This study may show that the use of PPI is not associated with the risk of HCC possibly because the study population consisted of participants with a relatively healthy liver. Future studies are warranted to better define individuals who may have harmful effects due to PPI use.

The strength of our study is a large number of study participants over 400,000 Koreans and precise drug prescription records based on a claims database that allowed precise evaluation of association of PPI with HCC. Second, we excluded participants with cirrhosis, hepatitis, and HCC before the index date, and adjusted potential confounders that may be associated with HCC risk to increase the accuracy of the results. Last, we stratified the participants according to the potential key variables for HCC development, such as CLD, GERD, aspirin, and NSAIDs, to clearly show the association between PPI use and HCC risk in selected populations. There are several limitations to be considered in this study. First, the NHIS database has no family history of HCC and dietary habits. Second, this study did not consider other acid suppressants such as histamine 2 receptor antagonists. Third, since this is a Korean population-based cohort study, we call for future studies based on other ethnic populations to validate our results for the generalization of the findings. Last, even though our study consists of a large number of study participants, the proportion of participants who took PPI is lower than that of those who did not take PPI. Therefore, the association between PPI use and HCC risk may need to be checked with a randomized controlled trial by making the two proportions similar in a future study. 

In conclusion, this retrospective nationally representative cohort study demonstrated that PPI is not associated with HCC risk. PPI users with potential confounders for HCC risk, such as obesity, CLD, and GERD were also not associated with increased HCC risk. Although no statistical significance was found, slightly increased and decreased risks of HCC were found according to PPI use in patients with old age and obesity, respectively. Future prospective studies are required to better confirm the association of PPI use with HCC risk in specified conditions, such as obesity.

## Figures and Tables

**Figure 1 jcm-11-02865-f001:**
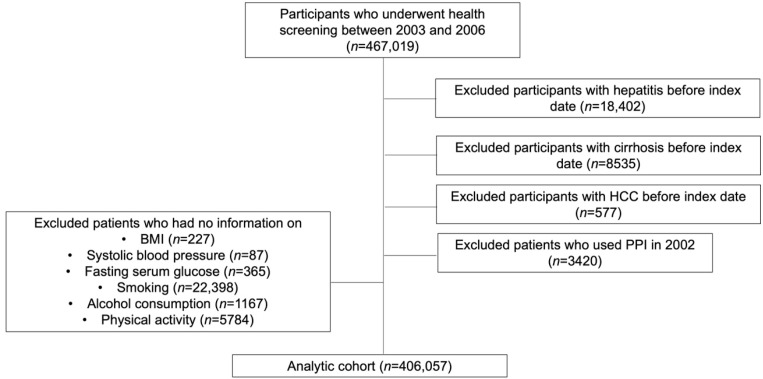
Study flow chart describing study selection.

**Figure 2 jcm-11-02865-f002:**
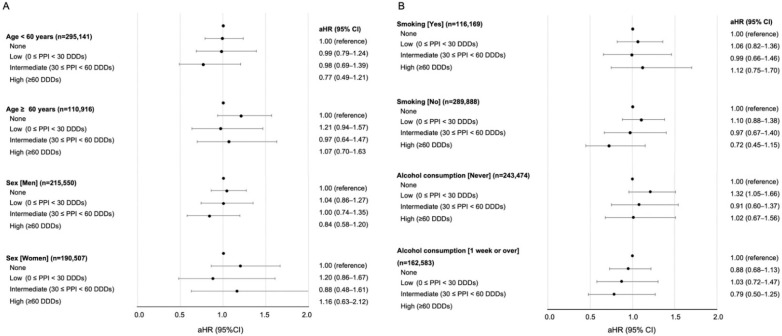
Stratified analyses on the association of PPI use and risk of HCC. Adjusted hazard ratio calculated after adjustments for age, sex, household income, body mass index, systolic blood pressure, fasting serum glucose, smoking, alcohol consumption, exercise frequency, and Charlson comorbidity index. (**A**) Stratified according to age and sex. (**B**) Stratified according to smoking and alcohol consumption.

**Table 1 jcm-11-02865-t001:** Characteristics of the participants who underwent health screening.

Characteristic	Participant (*n* = 406,057)
Age, years	51 (45–60)
Sex, *n* (%)	
Men	215,550 (53.1)
Women	190,507 (46.9)
Household income, *n* (%)	
First quartile	64,001 (15.8)
Second quartile	86,612 (21.3)
Third quartile	117,440 (28.9)
Fourth quartile (highest)	138,004 (34.0)
Charlson comorbidity index, *n* (%)	
0	179,113 (44.1)
1	123,439 (30.4)
≥2	103,505 (25.5)
Systolic blood pressure, mmHg	126 (114–138)
Diastolic blood pressure, mmHg	80 (70–85)
Body mass index, kg/m^2^	23.9 (22.0–25.8)
Fasting serum glucose, mg/dL	93 (84–104)
Cigarette smoking, *n* (%)	
Never	289,888 (71.4)
Past	34,783 (8.6)
Current	81,386 (20.0)
Alcohol consumption frequency, *n* (%)	
0	243,474 (60.0)
2–3/month	57,115 (14.1)
1–2/week	64,122 (15.8)
2–3/week	25,775 (6.3)
≥5/week	15,571 (3.8)
MVPA, time/week, *n* (%)	
0	213,675 (52.6)
1–2	103,074 (25.4)
3–4	46,404 (11.4)
≥5	42,904 (10.6)

Data are the median (interquartile range) unless indicated otherwise. Acronyms: MVPA, moderate-to-vigorous physical activity.

**Table 2 jcm-11-02865-t002:** Association of proton pump inhibitor use with risk of HCC according to DDDs.

	None	Low (0 ≤ PPI < 30 DDDs)	Intermediate (30 ≤ PPI < 60 DDDs)	High (≥60 DDDs)	*p* for Trend
Participant, *n*	341,734	38,340	14,776	11,207	
Overall HCC					
Event (%)	1215 (0.4)	153 (0.4)	56 (0.4)	42 (0.4)	
Person-year	2,334,893	261,565	100,381	75,668	
Crude rate/10,000 PY	5.2	5.8	5.6	5.6	
HR (95% CI)	1.00 (reference)	1.12 (0.95–1.33)	1.07 (0.82–1.40)	1.07 (0.78–1.45)	0.546
aHR (95% CI) ^a^	1.00 (reference)	1.12 (0.94–1.32)	1.00 (0.77–1.31)	0.92 (0.67–1.25)	0.559
aHR (95% CI) ^b^	1.00 (reference)	1.07 (0.91–1.27)	0.96 (0.73–1.26)	0.86 (0.63–1.17)	0.613
Cirrhosis-associated HCC					
Event (%)	236 (0.1)	27 (0.1)	8 (0.1)	11 (0.1)	
Person-year	2,330,710	260,987	100,225	75,352	
Crude rate/10,000 PY	1.0	1.0	0.8	1.5	
HR (95% CI)	1.00 (reference)	1.02 (0.69–1.52)	0.79 (0.39–1.60)	1.44 (0.79–2.79)	0.594
aHR (95% CI) ^a^	1.00 (reference)	1.02 (0.68–1.51)	0.74 (0.36–1.49)	1.25 (0.68–2.28)	0.731
aHR (95% CI) ^b^	1.00 (reference)	1.01 (0.68–1.51)	0.73 (0.36–1.48)	1.20 (0.65–2.22)	0.762
Non-cirrhosis-associated HCC					
Event (%)	979 (0.3)	126 (0.3)	48 (0.3)	31 (0.3)	
Person-year	2,335,303	261,611	100,401	75,690	
Crude rate/10,000 PY	4.2	4.8	4.8	4.1	
HR (95% CI)	1.00 (reference)	1.15 (0.95–1.38)	1.14 (0.85–1.52)	0.98 (0.68–1.40)	0.420
aHR (95% CI) ^a^	1.00 (reference)	1.14 (0.95–1.37)	1.06 (0.80–1.42)	0.84 (0.59–1.20)	0.364
aHR (95% CI) ^b^	1.00 (reference)	1.09 (0.90–1.31)	1.01 (0.76–1.53)	0.78 (0.54–1.12)	0.424

HR calculated using the Cox proportional hazards regression. ^a^ Adjusted for age and sex. ^b^ Adjusted for age, sex, household income, body mass index, systolic blood pressure, fasting serum glucose, smoking, alcohol consumption, exercise frequency, and Charlson comorbidity index. Acronyms: DDD, defined daily dose; HCC, hepatocellular carcinoma; PY, person-years; HR, hazard ratio; aHR, adjusted hazard ratio; CI, confidence interval.

## Data Availability

The datasets generated and/or analyzed during the current study are available from the Korean National Health Insurance Service, https://nhiss.nhis.or.kr (accessed on 1 January 2022).

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
