# Peer review of "Association between Proton Pump Inhibitor Use and Risk of Hepatocellular Carcinoma: A Korean Nationally Representative Cohort Study"

_jcm, 2022, doi:10.3390/jcm11102865_

Round 1

Reviewer 1 Report

The authors have performed a retrospective analysis of patients using Proton Pump Inhibitors and the incidence of HCC among this cohort.

The study cohort consists of a large number of patients. However, the proportion of patients who received PPI is low compared to the total cohort. Therefore, I am not sure if the conclusions from this study is skewed by various other factors unrelated to PPI usage.

The study does not discuss any mechanistic insights relevant to PPI usage and HCC incidence.

Considering that there have been conflicting reports about the correlation between PPI usage and HCC occurrence, could there be a regional aspect to this correlation? Could it be possible that people belonging to certain ethnicities be more susceptible to HCC incidence upon PPI usage?

These are some of the questions that would be great, if addressed in the paper.

It would be great if the authors used some figures to represent some of the data, in addition to the tables. Also, it would be great if the authors could explain more about the statistical analysis that was done, and why that specific test was chosen.

Reviewer 2 Report

The article "Association between Proton Pump Inhibitor Use and Risk of Hepatocellular Carcinoma: A Korean nationally representative cohort study" is an interesting and tough work on a controversial subject, the link between HCC and the use of PPIs. The authors performed a cohort retrospective study on an important number of patients and explained on a flow chart the selection of patients included in this study.

The initial data of the participants who underwent a health screening, as well as the results are presented and discussed in three tables. The final conclusions are presented and compared with other data from the literature, requiring the need for more studies worldwide. The article is not loaded with unnecessary information and provides a survey on PPIs applied to a specific Korean population.

However, as minor problems, I would suggest the following corrections:

  1. To state more explicitly that the conclusions cannot be generalized, especially those relating to obesity, old age, etc.
  2. To present more accurately the data ranges, for example:

- Instead of (> 0 and <30 DDD),

I would recommend (0 <PPI <30 DDD).

- Instead of (≥30 and <60 DDD),

I would recommend (30 ≤ IPP <60 DDD) and so on.

  1. A final table with Abbreviations would be very welcome.
  2. References should be presented throughout the article more accurately, followed by a full stop, and NOT vice versa.

So YES, that's right                       [12, 13].

So NO, that's wrong                     . [12,13]

More English and style issues could be improved by a native English speaker at a final reading.

Overall, I congratulate the authors for preparing this article, which required a lot of work.
